# AGENTIC ORCHESTRATION OF DRUG DISCOVERY ML TOOLS UNDER PARTIAL OBSERVABILITY

**Sara Masarone**
Ignota Labs
Cowley Rd, Milton, Cambridge CB4 0WS
`sara.masarone@ignotalabs.ai`

**Katie V. Beckwith**
Ignota Labs
Cowley Rd, Milton, Cambridge CB4 0WS
`katie.beckwith@ignotalabs.ai`

**Matthew Mason**
Ignota Labs
Cowley Rd, Milton, Cambridge CB4 0WS
`matthew.mason@ignotalabs.ai`

**Thomas Clelford**
Ignota Labs
Cowley Rd, Milton, Cambridge CB4 0WS
`thomas.clelford@ignotalabs.ai`

**Arran Willmott**
Ignota Labs
Cowley Rd, Milton, Cambridge CB4 0WS
`arran.willmott@ignotalabs.ai`

**Layla Hosseini-Gerami**
Ignota Labs
Cowley Rd, Milton, Cambridge CB4 0WS
`layla.gerami@ignotalabs.ai`

## ABSTRACT

Over 90% of drugs fail in clinical trials, often due to unanticipated safety issues, resulting in substantial financial losses and missed therapeutic opportunities. We develop models that uncover mechanistic bases of observed safety liabilities and support rational drug modification, including an asset-sourcing agent, a cheminformatics module predicting off-target interactions, and a bioinformatics module linking off-targets to toxicity pathways.

Individually, these tools are performant but outputs are fragmented, and not all input data are always available, limiting rapid decision-making. To address this, we introduce an orchestrating agent that dynamically coordinates tool execution based on data availability, task context, and uncertainty. The agent selectively invokes, sequences, or defers modules to enable adaptive analysis under partial information. We present its architecture and early testing, illustrating a framework to unify a fragmented AI ecosystem into a coherent, agent-driven system.

## 1 INTRODUCTION

Over 90% of drugs fail during clinical trials Arrowsmith & Miller (2013). Many of these failures involve compounds that may have otherwise been effective but were terminated due to toxicity signals that are complex and context-dependent. Recovering or repurposing such compounds represents a major opportunity to reduce attrition and accelerate the development of promising therapeutics Al Khzem & Wali (2025).

Computational approaches, including bioinformatics, cheminformatics, and machine learning, have enabled increasingly detailed modelling of compound behavior, such as off-target interactions and pathway perturbation. These tools are often trained on large, domain-specific datasets and achieve strong performance within their respective scopes Trapotsi et al. (2022). As a result, contemporary discovery pipelines are no longer limited by the availability of predictive models, but rather by the challenge of integrating multiple heterogeneous datasets and tools, each with distinct assumptions, input requirements, and output representations into a coherent decision-making process Seal et al. (2025).

We present an early-stage orchestrating system that coordinates heterogeneous predictive models under uncertainty and partial observability. This framework provides a path toward adaptive, agent-

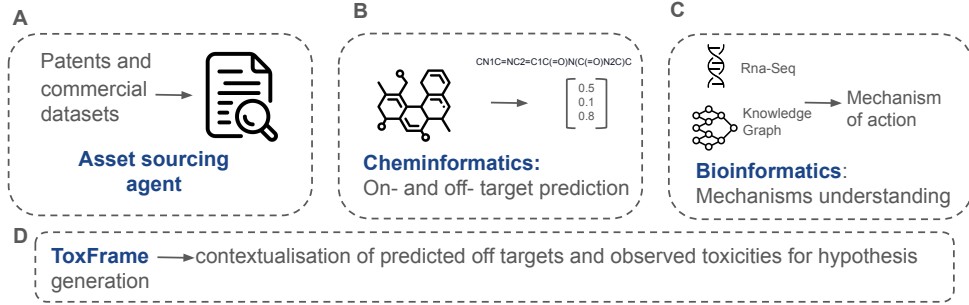

Figure 1: The schematic illustrates the tools we use in our platform. These include an asset sourcing agent that uses patents and commercial datasets to identify leads (A), a cheminformatics module capable of predicting on- and off- targets (B), a bioinformatics module that uses RNA-Seq data to identify plausible hypotheses for the mechanism of action (C) and ToxFrame, a contextualising tool linking predicted off targets to observed toxicities (D).

driven reasoning for drug turnaround, demonstrating how integrating complementary models can uncover actionable insights from complex, incomplete data.

## 1.1 TOOLS AND SYSTEMS

Our drug rescue platform integrates specialised bioinformatics and cheminformatics tools that have been independently validated across multiple projects. The platform is designed to support rapid identification of drug candidates with safety liabilities and the generation of mechanistic hypotheses to guide rational modification (illustrated in Figure 1).

The platform includes several core modules:

- Asset-sourcing agent (Appendix A.1): integrates commercial datasets, scientific literature, patents, and structured databases at a scale and speed beyond human capacity. Produces reports combining molecular structures, experimental findings, pharmacokinetics, and clinical side-effect profiles.
- Cheminformatics module (Appendix A.2): predicts on- and off-targets from molecular structures, encompassing over 15,000 models and leveraging one of the largest mitochondrial toxicity datasets ($>$50,000 data points).
- Bioinformatics causal reasoning engine (Appendix A.3): links predicted molecular targets to downstream transcriptional changes to generate mechanistic pathway hypotheses.
- ToxFrame (Appendix A.4): a literature-grounding tool that contextualises predicted off-targets and links them to observed toxicities, ranking hypotheses by evidence and flagging gaps for further investigation.

Currently, these tools are operated in a human-in-the-loop workflow. Expert scientists must determine which analyses to run, integrate outputs across modules, and consult literature, databases, or Google to contextualise results. While effective, this process is time-consuming, fragmented, and difficult to scale: even experienced teams spend days synthesising insights for a single asset.

## 2 PROBLEM SETTING

To manage an increasing volume of internal leads, we are developing an orchestrating agent designed to evaluate analysis requirements and autonomously trigger the appropriate platform modules, as illustrated in figure 2.

A primary limitation of traditional bioinformatics and cheminformatics pipelines is their rigid assumption that all required input modalities are available at the outset of analysis. In real-world asset

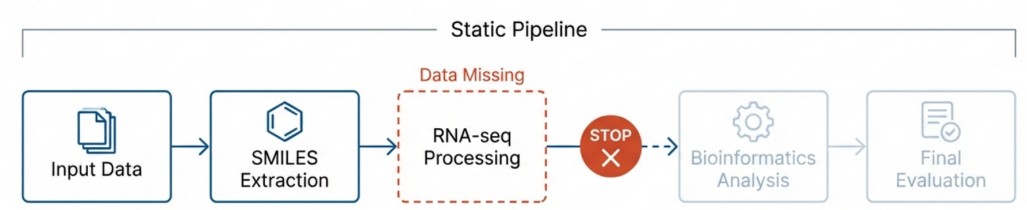

Figure 2: Diagram showing how linear pipelines may fail in the absence of required data.

evaluation, this assumption rarely holds. Data are often fragmented or incomplete, leading to partial observability over molecular structures, transcriptional responses, or mechanistic context. Static, linear workflows frequently fail under these conditions, halting analysis or producing incomplete results. In practice, human experts must step in to fill gaps, manually gather missing data, run tools in the right order, integrate outputs, and consult literature or databases to interpret results. This is slow, repetitive, and difficult to scale.

Our dynamic orchestration architecture addresses these limitations. Rather than relying on humans to manage workflows, the agent autonomously determines which modules to run, in which order, and with what inputs, executing the full workflow end-to-end and generating a consolidated report for the human expert. This allows scientists to focus their cognitive effort where it adds the most value, interpreting integrated mechanistic hypotheses and prioritising follow-up experiments, while the agent handles execution, data integration, and preliminary analysis.

## 2.1 MANAGING MISSING MODALITIES

Figure 2 illustrates how rigid, linear pipelines fail in the absence of required modalities, motivating the need for dynamic orchestration. The orchestration layer specifically manages the missing modalities that often halt standard workflows:

- SMILES Acquisition: On- and off-target prediction requires machine-readable molecular representations (SMILES) that are frequently unavailable for external or poorly documented assets Weininger (2002). When SMILES are missing, the agent determines whether structures can be retrieved from internal databases or whether auxiliary tools should be invoked to derive the representation prior to cheminformatics analysis

- Transcriptional Data: Bioinformatics causal reasoning relies on RNA-seq data, which is often absent due to experimental cost or time constraints. When transcriptional data are unavailable, the agent recognises this limitation and uses external databases e.g., Reactome to annotate pathways for predicted targets and maximise insight from molecular and prior knowledge, rather than terminating the analysis Milacic et al. (2024).

By explicitly managing missing modalities such as SMILES and transcriptional data, the orchestration layer ensures that analyses can proceed without interruption, even under partial observability. To evaluate this capability, we conducted preliminary testing to examine the agent's ability to maintain workflow continuity and generate mechanistic hypotheses under both full-data and partial-data scenarios.

## 3 SET UP AND PRELIMINARY TESTING

We present preliminary results from early-stage testing of the orchestration layer for two drugs which cause Drug Induced Liver Injury (DILI). Figure 3 shows two example: a workflow including all the necessary input data and a workflow missing RNA-Seq data.

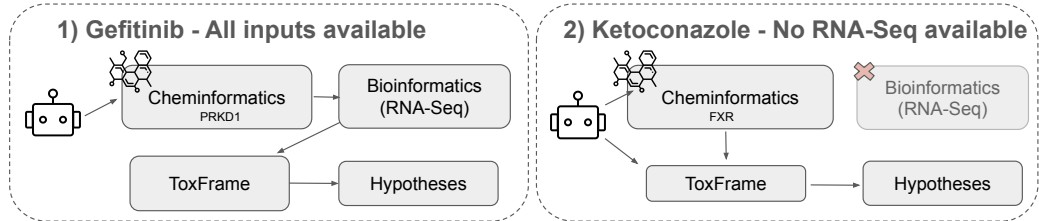

Figure 3: Preliminary testing of our orchestrating agent showing the coordination of a workflow with all the input data (1) and with partial input data (2).

## 3.1 UNCOVERING GEFITINIB'S MECHANISMS OF HEPATOTOXICITY

Gefitinib is an efficacious treatment for non-small cell lung cancer (NSCLC) but is classified by the FDA as Most-DILI-Concern due to severe hepatotoxicity. We used Gefitinib as a test case for our orchestrating agent, as all required input data were available included treated hepatocytes (liver cells), allowing execution of the full internal toolchain.

The agent autonomously coordinated the following workflow:

Gefitinib → Can we find SMILES? → Yes → Run the cheminformatics module → Generate on- and off-targets → Do we have transcriptomics data? → Yes → Run Bioinformatics module → Run ToxFrame for further contextualisation → Gather results and generate hypotheses.

PRKD1 was predicted as a novel off-target, causally linked to modulation of SPHK2 and dysregulation of genes involved in sphingolipid metabolism. ToxFrame recovered supporting biological context, including established links between PRKD1 and sphingolipid metabolism (Reactome), associations between sphingolipid dysregulation and liver toxicity, and reports of hepatotoxicity for other PRKD1 inhibitors Milacic et al. (2024); Li et al. (2020). This evidence supported the hypothesis Gefitinib → PRKD1 → sphingolipid metabolism disruption → liver toxicity, which was scored 8/10 based on literature strength, with the key uncertainty being direct Gefitinib–PRKD1 binding.

To test this gap, radiometric HotSpot™ kinase profiling confirmed previously unreported off-target activity of Gefitinib against PRKD1. This demonstrates that our system can move from data integration to mechanistic, testable toxicity hypotheses and identify actionable off-target liabilities.

## 3.2 KETOCONAZOLE - NO TRANSCRIPTIONAL DATA

Our second case study examines Ketoconazole, a well-characterised Drug-Induced Liver Injury (DILI) compound for which no RNA-Seq data were available. This scenario tested the agent's ability to detect missing transcriptional inputs and dynamically re-route its workflow toward structure- and literature-driven analysis.

Upon identifying the data gap, the agent prioritised target prediction and knowledge-base interrogation. FXR (NR1H4) was identified as a key functional target. Literature evidence linked FXR modulation to cholestasis, a known clinical manifestation of Ketoconazole-induced liver injury, and Reactome pathways further contextualised FXR's role in hepatic bile acid homeostasis Jackson & Brouwer (2025); Norona et al. (2020). Independent reports of Ketoconazole-associated cholestasis provided direct phenotypic support for this mechanism Velayudham & Farrell (2003).

This resulted in the mechanistic hypothesis: Ketoconazole → FXR dysregulation → bile acid metabolism disruption → cholestatic DILI, derived without reliance on transcriptomics data.

## 3.3 CONCLUSION

Although these results represent early-stage testing, they provide a preliminary proof of concept for our orchestrating agent's ability to navigate the partial observability inherent in real-world drug discovery. The successful identification and subsequent lab validation of Gefitinib's off-target activity against PRKD1 illustrates how autonomous orchestration can uncover actionable biological insights

from incomplete datasets. Future work will focus on further "closing the loop" by enabling the agent to autonomously prioritise and trigger the very wet-lab experiments it currently identifies, fostering a truly lab-in-the-loop scientific ecosystem.

## 4 MEANINGFULNESS STATEMENT

Our submission fits the Learning Meaningful Representations of Life workshop by presenting an application of multimodal representation learning to drug safety. We introduce an autonomous agent that orchestrates diverse cheminformatics and bioinformatics tools to interpret complex biological data. By integrating chemical structures with systems biology and omics data, our work demonstrates how learned representations can yield actionable scientific insight into drug toxicity. This approach aligns with the workshop's focus on modelling biology across scales and provides a framework for interpretable, mechanistic discovery. Ultimately, our research showcases an impactful real-world application of AI in understanding and predicting adverse biological phenotypes.

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

# A APPENDIX

## A.1 ASSET SOURCING SYSTEM

Our asset sourcing system searches clinical trial databases e.g., clinicaltrials.gov then applies automated screening to rank candidates against investment criteria. Users can deploy autonomous research agents to conduct deep research on selected assets, generating comprehensive reports that cover development history, target biology, and clinical trial outcomes. This automation reduces research time to around 10 minutes per drug; dramatically scaling the team's capacity to identify and evaluate potential acquisition targets.

The pipeline operates in a two-stage process. Stage one is a fast set of heuristic checks designed to filter the list of potential targets down to a tractable number. Stage two is a set of deep research agents which combine public and commercial data sources to produce a detailed dossier on the drug, the target, and the observed safety problems.

The heuristics, such as whether a drug is still in active development, are presented to the human operator to allow them to decide whether to progress the drug for deeper research. This process is designed to be lightweight, fast, and parallelisable. It can process tens of thousands of drugs in an hour and returns to the human a list of drugs found with the reasons why they may/may not be suitable for deeper research. To keep this process fast, most of these heuristics are rules-based. An exception is the component that reads the details of the clinical trial to decide whether or not treatment-emergent adverse events were present. This uses a fast LLM to parse the text and return structured output. We have used evals to iteratively improve and benchmark the prompt for this component to minimise incorrect results.

The user can then trigger AI research agents to compile a detailed dossier on each drug of interest. The research report contains sections on the drug, its target, and the state of its clinical development. These deep researchers use Tavily to search through clinical trial results, press releases, and academic papers to find up-to-date and relevant information.

Research reports are passed through a fact-checking module which extracts individual claims from the report and analyse the sources to ensure that the content of the report is well-supported and accurate. Finally an executive summary is produced to give researchers an at-a-glance understanding of the drug programme of interest.

## A.2 CHEMINFORMATICS MODULE

Our cheminformatics module uses a proprietary architecture to predict on- and off-targets for over 5,000 proteins at four therapeutically relevant concentrations. This is a highly effective in-silico tool, screening far more proteins than possible with wet-lab experiments. The architecture overcomes common challenges in chemical datasets, such as their limited size and structural diversity, by independently processing and fusing three distinct molecular representations: topological fingerprints, atomic graph structures, and large-scale, generalised chemical priors derived from a foundation model (ChemGPT) Frey et al. (2022). This multi-modal approach results in superior predictive performance, demonstrating an improvement of 7.73% in F1 score and 10.63% in balanced accuracy over the next best baseline across multiple challenging bioactivity and toxicity datasets (namely BBP, BACE, HIV, TOX21, ClinTox, MitoTox, CaV1.2 and NaV1.5).

## A.3 BIOINFORMATICS MODULE

Our bioinformatics module is a causal reasoning engine that connects molecular targets to downstream transcriptional changes by reasoning over a knowledge graph of known biomolecular interactions. Given input transcriptional data (such as RNA-seq), it optimises a subnetwork that represents how off-target binding drives the observed changes in gene expression.

The platform works by first using differentially expressed genes to derive transcription factor activities, which form the bottom layer of the subnetwork.

These transcription factors are then connected to molecular (off) targets based on the on-and off target predictions at the top layer of the subnetwork, by reasoning over a knowledge graph of protein-

protein, drug-protein, drug-side effect, and protein-side effect interactions. This network can either be a global interactome or constrained to a specific tissue or cell type of interest by using basal protein expression levels to exclude or include a particular protein.

The reasoning is based on integer linear programming (ILP) optimisation, where the objective function aims to optimise a network in which the node activities explain the observed transcription factor activities, while minimising the total number of nodes in the network.

The resulting sub-network is further analysed with pathway enrichment to understand the active pathways on which the compound is acting, enabling mechanistic understanding of what a drug does at the cellular level, and how the predicted off-target could feasibly cause the observed biological response. The diagram below (A.3 illustrates the bioinformatics module in more details)

### A.4 CONTEXTUALISATION WITH TOXFRAME

ToxFrame is an internal hypothesis-generation and evidence-prioritisation tool designed to contextualise the cheminformatics outputs (on and off target preductions) supporting early research decisions in target discovery and safety assessment. Its primary function is to reduce uncertainty around predicted on- and off-target interactions by systematically identifying, validating, and summarising relevant external evidence. The tool operates across both adverse-event-agnostic and side-effect-specific research modes, allowing teams to either broadly assess biological targets and pathways for potential liabilities or focus directly on a defined safety signal. ToxFrame integrates automated literature and web-based data extraction with structured evidence evaluation, enabling research teams to focus on the most credible, biologically meaningful, and contextually relevant hypotheses.

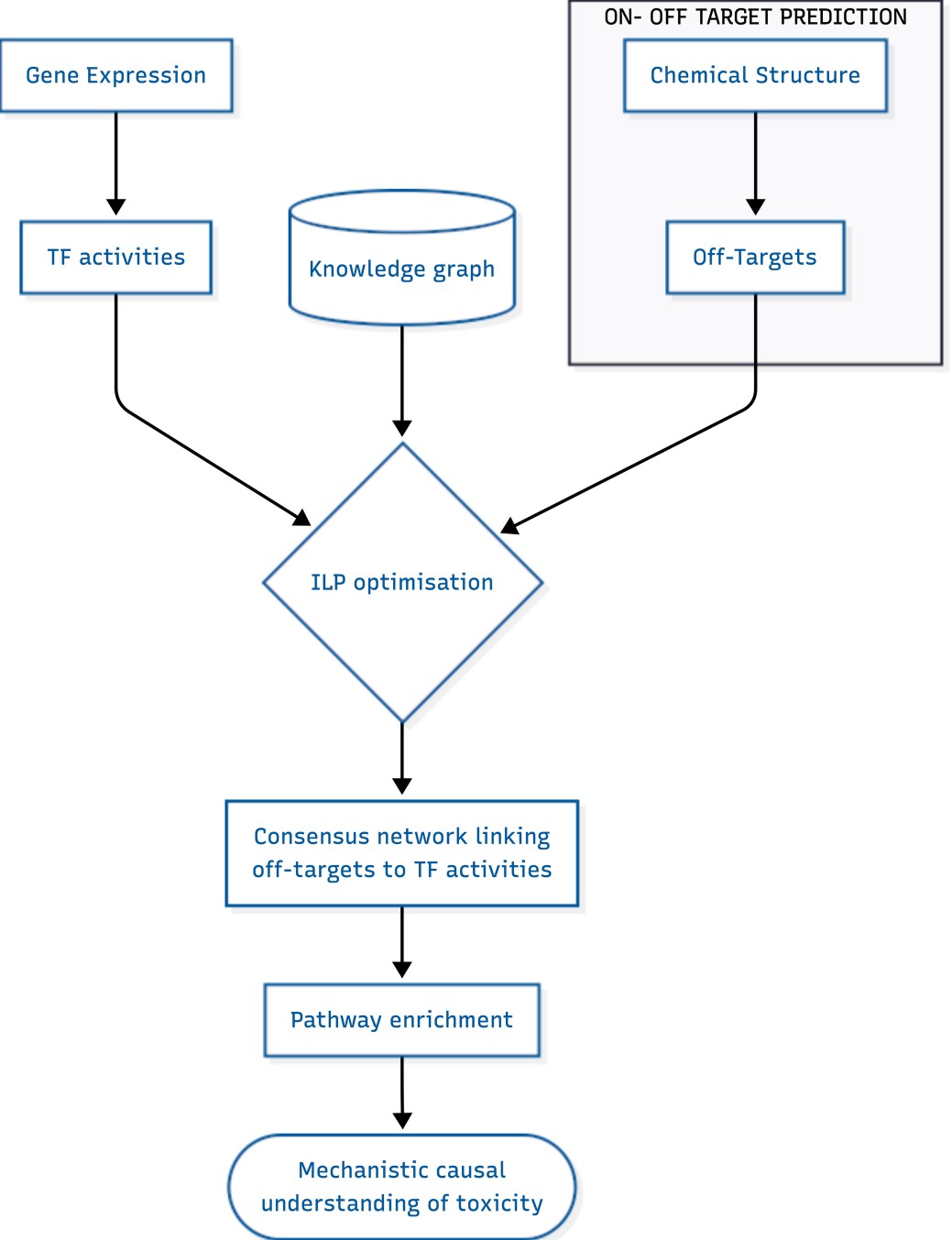

Figure 4: Diagram showing the bioinformatics pipeline in detail.

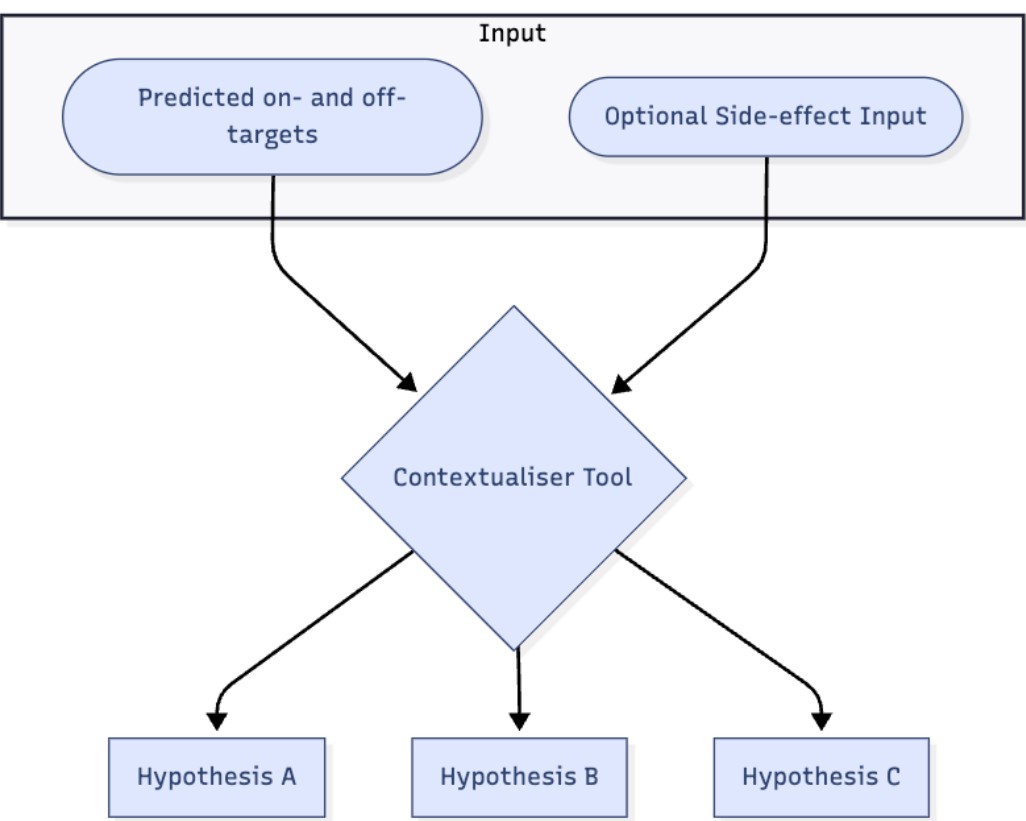

Figure 5: Diagram showing the bioinformatics pipeline in detail.

