# OpenReview forum: "Agentic orchestration of drug discovery ML tools under partial observability"
_ICLR.cc/2026/Workshop/LMRL — ICLR 2026 Workshop LMRL Poster_

### Official Review · Reviewer_eyaW · 2026-02-18
**Good start on a promising idea, would love to see developed further.**

**Rating:** 7
**Confidence:** 4

**Review:**

# Quality
## Pros
- Well-written, no obvious typos or grammatical issues.
## Cons
- Fairly sure that the abstract is supposed to be just one paragraph.
- Would love to have seen at least one quantitative evaluation of the orchestration system.

# Clarity
- Very clear to me how the dynamic orchestration works, and what each module does.

# Originality
- Does not seem particularly original, except in its use of a dynamic orchestration agent, which seems like an important but common-sense modification. I am not in a position to say, however, how novel this really is. Perhaps the paper should play up more the importance and difficulty of being dynamic.

# Significance
- Without quantitative evaluation and comparisons, it is difficult to say whether this is a significant contribution or not. However, I think that ideas like this certainly have a significant role to play in future drug discovery workflows, and making them adaptive to present or missing modalities is certainly crucial in their performance.
- It is not clear to me how related to representation learning for cells, the putative focus of this workshop, but this work is I suppose at least tangentially related in that it uses AI to combine different modalities to make biological predictions.

---

### Meta-Review · Area_Chair_9ffK · 2026-02-28

**Recommendation:** Accept (Poster)
**Confidence:** 4

**Metareview:**

This paper unfortunately only received a single review, but having looked at the paper myself, I agree that it's worth discussing at LMLR: agentic workflows for these tasks are increasingly prevalent and I think LMLR is a good venue for discussing their strengths and limitations.

---

### Decision · Program_Chairs · 2026-03-02

**Decision:**

Accept (Poster)

**Comment:**

Please see the meta-review.